# Under-Reporting of Tuberculosis Disease among Children and Adolescents in Low and Middle-Income Countries: A Systematic Review

**DOI:** 10.3390/tropicalmed8060300

**Published:** 2023-05-31

**Authors:** Alexandra R. Linn, Melanie M. Dubois, Andrew P. Steenhoff

**Affiliations:** 1Global Health Center, Children’s Hospital of Philadelphia & Department of Pediatrics, University of Pennsylvania, Philadelphia, PA 19104, USA; 2Department of Internal Medicine, Hospital of the University of Pennsylvania, Philadelphia, PA 19104, USA; 3Division of Pediatric Infectious Diseases, Weill Cornell Medicine, New York, NY 10065, USA

**Keywords:** TB, reporting gap, youth, public health, child, adolescent, pediatric, childhood

## Abstract

Under-reporting of tuberculosis (TB) disease in children and adolescents is a significant global concern, as many children are missing from TB notification data. A systematic literature review was conducted to understand the global reporting gap of child and adolescent TB as well as current interventions to close this gap in Low- and Middle- Income Countries (LMIC). Our study found large and variable gaps in child and adolescent TB reporting, due to various factors. Interventions to close this gap exist but are limited. Future studies are necessary to improve global surveillance systems to improve TB care delivery for children and adolescents.

## 1. Introduction

In 2021, there were an estimated 10.6 million people who contracted tuberculosis (TB), with children accounting for 11% of cases [1]. Yet there are likely many children missing from national TB surveillance systems, particularly as data gaps exist between the reported number of children with TB and the estimated TB incidence [2,3]. The COVID-19 pandemic has additionally had a profound impact on childhood TB services, with decreased notifications particularly affecting the youngest children [4].

The World Health Organization (WHO) Onion model is a helpful framework for assessing the fraction of TB cases accounted for in TB notification data [5]. There are many steps involved in obtaining TB case notification data, which involves the continuum from identifying and diagnosing TB cases to reporting of TB cases to the national TB program. Additional factors such as access to healthcare limit the ability of patients to engage with the system, a challenge that can be profound in pediatric populations. TB is challenging to diagnose in children, given that children often present with varied, at times subtle clinical symptoms, usually cannot spontaneously produce sputum for assessment, and have paucibacillary disease [6]. Children are more likely to be missed in notification data since TB reporting relies on bacteriological confirmation in many settings. Once a child is diagnosed with TB, challenges exist in recording and reporting the case to the national TB program. One such barrier is that many children are referred to hospitals for TB evaluation and diagnosis, particularly those with severe disease, and then referred back to clinics to complete TB treatment, which is an opportunity for gaps in the recording and reporting of TB [7].

Countries such as Botswana, which found that a third of child TB cases were missing, have found surprisingly large gaps in reporting pediatric TB to national registers [7]. In high-burden settings, children may be diagnosed but never registered or reported to the national registration system, leading to incomplete treatment, underestimates of case numbers, and fewer resources allocated than are required. Our primary objective was to perform a systematic review to quantify the gaps in reporting for child and adolescent TB in low- and middle-income countries (LMIC). Our secondary objective was to describe interventions to help close the gap in reporting TB in children and adolescents.

## 2. Methods

### 2.1. Study Design

This study was a systematic review of the English literature to quantify under-reporting of pediatric and adolescent tuberculosis disease. The review adhered to the Preferred Reporting Items Systematic Reviews and Meta-Analyses (PRISMA) guidelines for systematic reviews [8,9].

### 2.2. Search Strategy

The search strategy was designed by two of the co-authors, AL and AS. It included different iterations of “child” and “pediatric” and “tuberculosis” and terms including “reporting”, “notification”, and “registration” to identify articles focusing on under-reporting. Search strategy was modeled off prior reviews on similar subject matter [10]. For this article, a reporting gap or under-reporting was defined as a case of tuberculosis disease that was diagnosed but not reported to a private or national TB surveillance system. Please see the full search strategy in Table 1.

We searched the PubMed and Ovid Databases between November 2022 and January 2023.

### 2.3. Inclusion and Exclusion Criteria

Articles were included if they were published in English in the past 30 years (1992–2022) and if they focused on LMICs, as defined by the World Bank [11]. Articles focusing on international, national, sub-national, or regional cohorts were included. Articles were included if they described pediatric and adolescent populations (0–24 years). Both articles summarizing stand-alone pediatric or adolescent cohorts as well as those nested within a complete lifespan cohort were included. While the WHO defines pediatric and adolescent populations as up to 19 years of age, we included ages up to 24 since multiple articles grouped adolescent populations in the 15–24-year age category [12]. In order to be included in the review, articles needed to either discuss TB under-reporting or the reporting gap and/or interventions to close the reporting gap. Articles were excluded if they only discussed a descriptive overview of TB cases or case notifications in a specific location without identifying a specific gap in TB reporting or an intervention to improve under-reporting. Studies were excluded if they focused on challenges with diagnosis or loss to follow-up rather than under-reporting.

All searched articles were uploaded to Covidence, an online platform for simplification of systematic reviews [13]. Duplicates were removed automatically by the Covidence system. Two reviewers, AL and MD, independently reviewed each abstract for inclusion which then moved into the next stage of full-text review. Discrepancies were resolved by a third reviewer, AS, prior to moving forward in the review. All full-text articles were reviewed, and inclusion was agreed upon by two reviewers, AL and MD. Discrepancies were again resolved by a third reviewer, AS, to determine final inclusion in the review. Two authors, AL and MD, hand-searched bibliographies of included works to identify additional articles for inclusion. The PRISMA flowchart of the systematic review can be found in Figure 1.

Data were then extracted to identify objectives, study design, reported gap, and/or intervention by two reviewers, AL and MD.

### 2.4. Quality Assessment

The quality of included articles was assessed using the Strengthening The Reporting of Observational Studies in Epidemiology (STROBE) guidelines [14]. All included full-text articles received a quality assessment by two reviewers, AL and MD. Studies were characterized as low (<33%), moderate (34–66%), or high (>66%) quality based on their ranking out of the 22 STROBE criteria. Of note, studies were still included if they were low or intermediate quality since reviewers felt these articles still made significant contributions to the review.

## 3. Results

Of the 506 records identified, 117 duplicates were removed, and 383 studies were screened. Forty-two were assessed for full-text eligibility. Eighteen articles were included in the final review with the most common reason for exclusion being the wrong study outcome. Details of the included articles are summarized in Table 2 and Table 3.

Of the included studies, 13 quantified the gap in reporting child and adolescent TB, and 5 described interventions to close this gap. The estimated gap in reporting ranged from 12% (54 children treated but unreported in the TB register/443 total children treated for TB in South Africa) to 98% (4746 child TB cases not recorded in the TB registers or reported to the National TB program/4821 total cases of child TB in Indonesia) and was described in eight LMICs including Benin, China, India, Indonesia, Kenya, Nigeria, Pakistan, and South Africa. A map showing the included countries with the reported quantified gap is shown in Figure 2.

Larger gaps in reporting were described in areas with higher disease burdens and in urban areas [15,16]. Patients with disseminated disease, more severe disease, and those under the age of five years were less likely to be reported [16,17,18,19]. Those diagnosed by and started on medicine by private providers were less likely to be reported to national TB registers [15,20]. Cases were less likely to be reported if death occurred prior to reporting/registration [17]. A clinical diagnosis was more likely to be under-reported than a bacteriologically confirmed case [21].

The number of effective interventions to improve the under-reporting of child and adolescent TB disease cases is limited. The included intervention studies sought to correct TB case under-reporting by strengthening linkage systems between facilities (referral hospitals and TB treatment clinics) [18,22], increasing communication following initial diagnosis [18], utilizing electronic systems and call centers for ongoing support [23], improving death registration [22], intensifying case finding with a direct linkage to registration [24], and engaging private providers [23,24].

In terms of quality, 16 articles met qualifications for high quality and two were characterized as moderate quality. The quality assessment scoring is included in Table A1. A few articles (three) shared efforts to decrease bias [25,26,27]. Two articles were unclear on their inclusion and exclusion criteria with minimal information about the age breakdown of participants [26,28].

## 4. Discussion

This is the first systematic review on this topic in the pediatric population. There is a wide range in the rates of under-reporting of pediatric and adolescent TB disease in LMICs coupled with a paucity of data from most countries. The review highlighted multiple trends in the under-reporting gap, which varied by location, age, clinical presentation, and healthcare system. Interventions to address under-reporting include improved clinical surveillance and case finding and strengthened linkages between hospitals and clinics where registration occurs. There remains limited data on potential interventions.

### 4.1. Reporting Gap

Our search strategy yielded only eight countries describing TB disease under-reporting in the pediatric population with South Africa contributing the majority of articles. The countries represented were predominantly in Africa and Asia with larger gaps seen in Asian countries.

In our included articles, patients were less likely to be reported in certain settings, including urban and high-burden TB areas [15,16,27]. Greater under-reporting in higher-burden areas may be the result of a larger patient load, inadequate staffing, and limited infrastructure to support the number of diagnoses made [15,27]. Prior articles have also noted misclassification in high-burden TB settings, given limited diagnostic tools, leading to both under- and over-diagnosis [29].

Private health systems also present a unique challenge. There were higher rates of under-reporting from the private sector reported in Kenya, India, and Pakistan [15,20,21,23,30]. Private practitioners often do not have direct links to national TB programs and thus may not provide data to NTPs. They frequently need to refer patients to TB clinics, but referral systems are weak [20]. Incentives, treatment access, and costs were brought up as challenges for private practitioners with fewer cases reported when private practitioners started treatment themselves as compared to referring children to the public system to commence TB treatment [20]. Methods of diagnosis can also differ. One article from Pakistan found that private practitioners were more likely to diagnose TB disease based on X-ray and clinical criteria rather than bacteriologically confirmed samples [20]. Education and standardization may also play a role with one study noting that private practitioners were less likely to utilize national TB guidelines for diagnosis [21].

Pediatric populations are uniquely vulnerable to under-reporting. In one included study from Indonesia, only 11 of 32 hospitals reported childhood cases of TB [25]. Younger ages, especially those less than five years, were less likely to be reported [25]. This general trend has been reported previously [31,32]. Articles theorize that practitioners feel less confident with the initial diagnosis and identification of TB in this age group, which leads to less confidence in reporting the cases to national TB registers. In Nigeria, results of healthcare provider interviews revealed that there was limited suspicion of TB in children, which could lead to missed diagnoses and limited recording of cases [26]. Young children are additionally more likely to have paucibacillary disease, which decreases the likelihood of culture-confirmed TB and increases the likelihood that they will be missed by microbiological surveillance [33]. There are variations in the implementation of contact tracing for young children, as one study found that fewer children were recorded in the TB treatment register in clinics without supervised contact tracing [16]. Other trends appeared to be unique to specific pediatric areas. For example, a study from Pakistan found higher levels of under-reporting in men than women whereas another study from South Africa did not find significant differences based on gender [17,21].

Patients were also less likely to be reported if they had extrapulmonary TB or drug-resistant TB [19,34]. In some locations, non-pulmonary forms of TB follow different diagnosis and treatment pathways. For example, one article from South Africa described how some children with TB meningitis are treated via a home-based program and may never present to a formal site for registration [17]. Another South African article described gaps in notifications for TB meningitis up to 44% [35]. An explanation for lower rates of reporting of extrapulmonary TB disease in children may be a lack of training and education about the identification of extrapulmonary TB in children among healthcare workers. Specifically, most healthcare worker TB training focuses on identifying smear-positive TB in adults [19]. One included study noted that patients with extrapulmonary tuberculosis may be prescribed medications directly through a pharmacy rather than through a DOT center leading to under-reporting [30]. Children with drug-resistant TB (DR-TB) can experience diagnostic delays and are often admitted to the hospital for several months of treatment, which can influence reporting of DR-TB [34].

Under-reporting was also more likely in children with severe disease and death [16,17]. Higher rates of under-reporting among children who die in the hospital were identified as an area of concern, as these patients are often unregistered and missing from global TB statistics [28]. An article from South Africa noted that ten children died prior to treatment referral, and none were recorded [17]. This can lead to an under-representation of the burden of diseases, an incomplete understanding of the causes of child mortality, and inadequate resource allocation in a given area.

Multiple papers discussed children with tuberculosis and HIV co-infections. Notable findings included that patients’ HIV status and antiretroviral therapy (ART) data were inconsistently reported and may not be recorded in specific databases such as laboratory registers [15,19]. A study from Benin showed that treatment success rates were lower in children with HIV and TB co-infection, which warrants improved monitoring of treatment outcomes in this sub-population [19]. One paper noted that patients with HIV co-infection were more likely to die while hospitalized or shortly after, which may lead to under-reporting [36]. Finally, when it came to treatment, one article from South Africa described how children with HIV were more likely to be fully treated for TB but classified as “not-TB” in the record due to challenges with diagnosis in this population. This highlights the possibility of misclassification in the TB registers for children living with HIV [16].

Additional gaps in TB recording are a result of the system of care delivery. In many settings, pediatric TB patients are diagnosed in hospitals, where there is no system of TB registration [27,28]. TB cases are referred for follow-up in outpatient clinics where they are registered and continued on treatment. Within this referral system, there are many patients who are missed due to the inability to present to local clinics, resulting in under-notification and loss to follow-up. In addition, there are gaps between the number of bacteriologically confirmed TB cases from laboratories and notifications in the TB treatment register [15,36]. Various components of these systems such as decentralized services can be strengthened to improve TB outcomes [37,38]. Hospital length of stay is one health system factor that can be considered within the system of care delivery. One article described an association between longer hospital stays and a lower likelihood of reporting. While no specific reason for this was investigated, the authors wondered whether children with more severe diseases were more likely to have longer hospital stays, and thus more likely to start treatment while admitted rather than present to clinics for treatment and registration [17]. Further studies should look into this association to help identify factors that may put children at higher risk for under-reporting.

Health systems are complex and multiple factors affect their performance including how well they report cases of TB disease in children and adolescents. The COVID-19 pandemic decreased TB notifications in children globally, as public health priorities shifted towards providing care for COVID-19 patients and away from childhood TB services [4]. Conflicts such as war and political instability often lead to decreased TB case notifications [39]. This warrants future research to identify resilient approaches to ensure childhood TB notifications are maintained during times of crisis, including pandemics and political conflicts.

Gaps in reporting of pediatric and adolescent tuberculosis are reported in both LMICs and non-LMICs, and reporting gaps need to be improved globally. For example, in Korea, 66.7% of children 0–14 and 40.6% of adolescents and young adults 15–29 were not reported to their national TB surveillance system [40]. In the UK, up to 20% of cases of tuberculosis in ages 1–15 may be missed in their surveillance network [31]. Similar gaps are seen when comparing our included articles to adult data in LMICs which is more prevalent in the literature [15,36,41,42,43,44,45]. Authors are also evaluating these gaps through mathematical modeling studies to further the understanding of under-reporting globally [3].

### 4.2. Interventions

Few evidence-based effective strategies exist to close this reporting gap. In general, interventions focused on different areas of the care cascade where gaps in reporting can occur including initial case finding, diagnosis, clinic linkage, and treatment follow-up. One study from Nepal found an association between a package of intensified case finding strategies and increased childhood TB case registrations [24]. A study from South Africa found that a simple hospital-based TB referral service improved the recording and reporting of childhood TB [18]. Another South African study utilized a TB Care Centre to bolster the referral and registration of all patients diagnosed with TB [22]. One study from South Africa looked at an intervention to incorporate complementary surveillance strategies, including clinical and laboratory surveillance [33]. A proposed solution from India is to educate healthcare providers and provide refresher training on the TB notification system and process for reporting [30].

Finally, a study from India included a mixed-methods approach to understanding the gap in reporting, which included interviews and focus group discussions with healthcare providers [30]. Several barriers and solutions were identified, including the need for improved communication, training, and hospital information systems to improve TB notifications [30]. This study design underscored the importance of having stakeholder voices and perspectives incorporated when designing an intervention. Other articles noted the importance of qualitative studies of this type to better understand the causes of under-reporting, especially in large hospital settings [27].

### 4.3. Limitations

Our study has several limitations. These include limited record-keeping and reporting linkages for childhood TB in high-burden settings [26,27]. There is additionally a lack of age and sex-disaggregated data for childhood TB [26]. Our study was limited to English language literature and only covered a 30-year period. Additionally, we limited our search to Pubmed, Medline, and references listed in papers. Hence, while this ensured high-quality English language papers as assessed by STROBE criteria, it also means we may have missed relevant data that have been reported in other languages or in the grey literature.

Our study maintained a narrow scope of quantifying initial under-reporting and did not include studies that focused on gaps and delays in treatment recording and registration, which is another important component of the care cascade [27]. Finally, we included studies of various sizes including regional, subnational, national, and international studies in our systematic review. Since some of these studies focus on gaps in specific locations and regions in various LMICs, they may have limited generalizability to describe national under-reporting in pediatric populations.

In addition, the adolescent population was not frequently referenced individually in our included studies. One study from Pakistan highlighted that the adolescent population is unique in potential means of spread in schools and in patients’ ability to produce sputum for testing when compared to younger populations [20]. Further studies focusing specifically on this population may identify additional opportunities for improvement in adolescent TB reporting.

### 4.4. Future Directions

Future studies can focus on various ways to improve reporting of childhood TB disease. It is critical to have accurate surveillance data to describe the epidemiology of global childhood TB and measure the impact of public health interventions, particularly in high-burden settings [16]. There is a need for increased supervision and investment in recording and reporting systems for childhood TB, with consideration of the use of technological innovations to help facilitate this [5,30]. While electronic records offer standardization and ease of reporting, they too have internal inconsistencies that require routine surveillance and improvement [34].

There is additionally a need to measure and report TB-related data at each step of the care cascade for children, including treatment and preventative therapy. Suboptimal contact tracing has been reported for young children, with opportunities for improvement in case detection and reporting in this high-risk group [16]. There are gaps in the delivery of pediatric TB medications, as one study described that only one childhood TB case received child-friendly anti-TB formulations, while the rest of the cases received a standard adult regimen [25]. Accurate data on the burden of childhood TB is necessary to drive action and planning for TB programs, particularly related to the market for pediatric medications [26]. Additional studies could assess novel approaches such as “hubs,” or facilities treating large numbers of childhood TB cases, to evaluate their ability to improve the reporting of childhood TB in LMIC at scale [26].

## 5. Conclusions

Under-reporting of pediatric and adolescent TB disease is a significant problem in LMICs. This under-reporting has implications including under-representation of the burden of TB disease and mortality in children and adolescents leading to inadequate resource allocation. Few evidence-based effective strategies exist to close this gap. Pragmatic solutions that effectively couple initiation of treatment with mandatory notification at all levels of care for every child and adolescent TB case are what is needed. Future research should assess the ability of practical, creative solutions to improve reporting accuracy with an emphasis on pediatric populations in high-TB burden countries.

## Figures and Tables

**Figure 1 tropicalmed-08-00300-f001:**
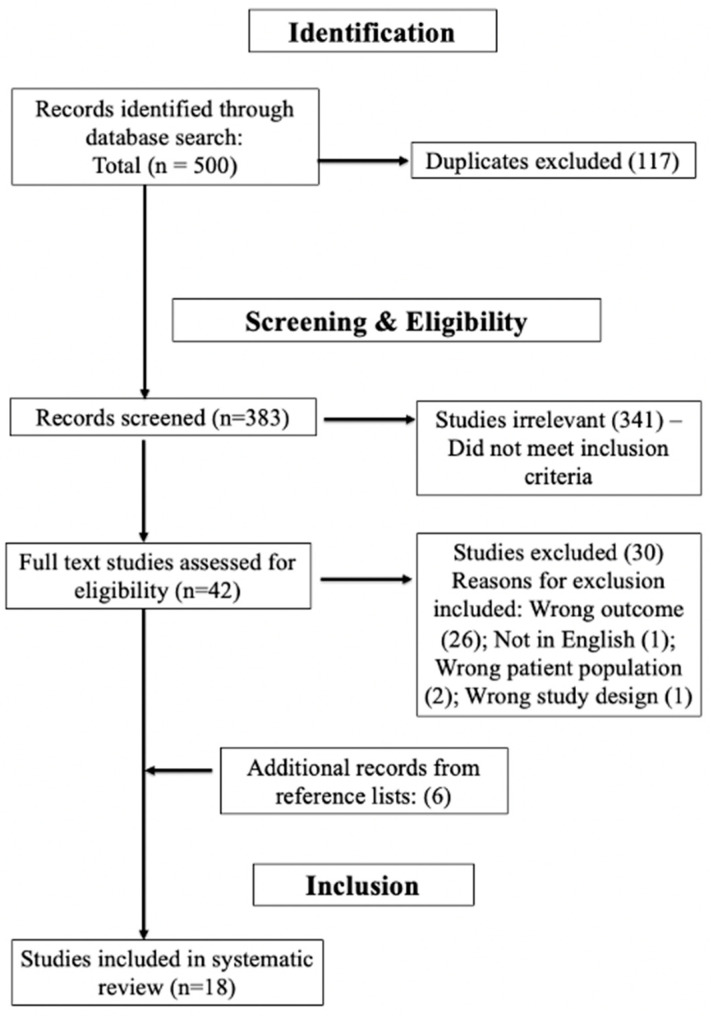
PRISMA flowchart.

**Figure 2 tropicalmed-08-00300-f002:**
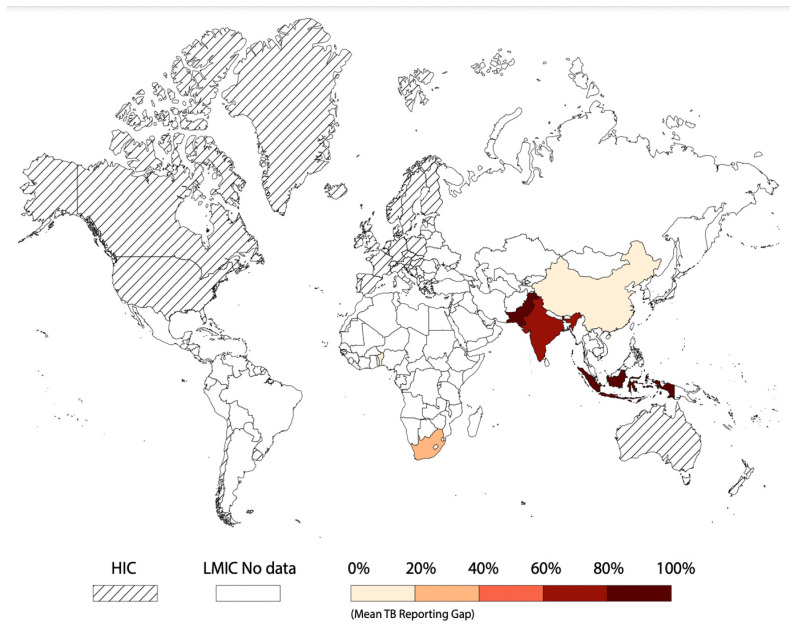
World map showing the magnitude of the reporting gap for child and adolescent TB in low- and middle-income countries.

**Table 1 tropicalmed-08-00300-t001:** Search strategy.

Set	PubMed/MedLine
1	Childhood tuberculosis
2	Childhood TB
3	Pediatric TB
4	Paediatric TB
5	Pediatric tuberculosis
6	Paediatric tuberculosis
7	Sets 1–6 were combined with “OR”
8	Under-reporting
9	Under reporting
10	Reporting gap
11	Registration
12	Notification
13	Sets 11–12 were combined with “OR”
14	Set 7 and Sets 8–10, 13 were combined with “AND”
15	Sets 1–14 were limited to 1992–2022

Note: Words were entered as free text when searched in the respective databases. When the search was performed, lower case was used for all search terms. These terms are capitalized in this table for clarity.

**Table 2 tropicalmed-08-00300-t002:** Included articles describing reporting gap.

Articles Describing Reporting Gap									
#	Paper	Study Type	Year	Country (ies)	Age Range (Years)	TB Type	Hospital Type/ Reporting Body	Reported Gap	Characteristics of Reported Gap
1	Berman et al.	Descriptive study	1992	South Africa	0–14	TB meningitis	75 hospitals in the Western Cape Health Region/ Department of National Health and Population Development	44% (105/238)	Noted that 16% of cases were excluded for an incorrect diagnosis, double notification, or other documentation errors.
2	Edginton et al.	Descriptive study including qualitative interviews	2005	South Africa	*	*	Chris Hani Baragwanath Hospital (tertiary hospital)/NTP	31% (285/1291)	Most patients who died were not recorded.
3	Marais et al.	Prospective Observational Study	2006	South Africa	0–12	General	5 primary healthcare clinics in Cape Town/NTP	12% (54/443)	Patients were less likely to be reported if diagnosed at a referral hospital or if had more severe disease.
4	du Preez et al.	Retrospective cohort study	2011	South Africa	0–12	General	Tygerberg Children’s Hospital in Cape Town/ETR.net	38% (101/267)	Patients were less likely to be reported if had disseminated TB (in sub-analysis if had TB meningitis, but not miliary TB) or death prior to referral.
5	Lestari et al.	Cross sectional study	2011	Indonesia	0–14	General	32 DOTS hospitals/NTP	98% (4746/ 4821)	More than half of child TB cases were in children less than 5 years old. Many DOTS hospitals did not have records of cases of child TB or report these cases to the NTP.
6	Rose et al.	Retrospective cohort study	2013	South Africa	0–14	General, drug- resistant	Tygerberg Children’s Hospital and its outreach clinics or Brooklyn Hospital for Chest Diseases (BHCD)/NTP	36% (28/77)	Only a small number of online registrations were children. If children were not referred to local TB hospitals or clinics, there was no mechanism to ensure they were registered in EDR.web.
7	Ade et al.	Cross- sectional, retrospective cohort study	2013	Benin	0–14	General	5 public or private basic management units (BMUs) in Cotonou/NTP	16% (29/182)	There was more under-reporting in children under 5 years old. Extrapulmonary TB was less likely to be reported, thought to be due to misdiagnosis by healthcare workers due to lack of training.
8	Coghlan et al.	Exploratory assessments, record reviews, interviews of healthcare providers	2015	Indonesia, Nigeria, Pakistan	*	*	Both public and private sector health facilities outside the network of national TB control programs (non-NTP facilities)/NTP	Varied; cases diagnosed but unreported— Indonesia (985), Pakistan (463), Nigeria (24) **	Private sector did not provide data to the NTP. There is a low level of suspicion for childhood TB in Nigeria generally.
9	Tollefson et al.	National-level retrospective TB inventory study	2016	Kenya	0–55+ (0–24)	General	Laboratory registers from public or private laboratories/ National TB surveillance systems (TIBU)	21% (715/3409)	Under-reporting was the greatest in the sub-counties with a high burden of TB, thought this may be due to pre-treatment loss to follow-up. Unreported cases were more likely if diagnosed at private and large facilities.
10	Li et al.	Retrospective inventory study (record review)	2018	China	<15, 15–64, ≥65	Pulmonary TB or TB pleurisy	Nine provinces across the eastern, central, and western regions of China/ Tuberculosis Information Management Systems (TBIMS)	19.3% (1082/ 5606)	Noted that age < 15 years, type of TB (pleurisy), recording source and region (eastern or central) were more likely to be under- reported. Discussed that large national and regional reference hospitals have high workload and limited resources for extra staff that affects reporting. The location where pediatric patients are typically treated (pediatric hospitals or large general hospitals) are not directly connected to TBIMS. Reporting regulations for TB pleurisy vary by province which can lead to under-reporting.
11	Fatima et al.	Nationwide cluster-based cross-sectional study	2019	Pakistan	0–14	General	Health facilities in 12 districts including NTP and non-NTP public health services, private health services, private laboratories/ NTP	78% (5070/ 6525)	Reporting differed by province; under-reporting was higher in boys (84%) than girls (68%). Under-reporting was more common in clinically diagnosed cases (78%) than bacteriologically confirmed cases (76%).
12	Siddaiah et al.	Mixed-methods study with retrospective review and key informant interviews and focus groups	2019	India	1–65+ (1–14, 15–24)	General	Private tertiary-level teaching hospital in Bengaluru, Karnataka State, South India/Indian RNTCP, a vertical national health program, with online notification portal Nikshay	76.8% (2935/ 3820) missing notifications, 9.3% (82/885) cases recorded in electronic portal For ages 0–14 (24/264, 9.1% notified) and for ages 15–24 (118/476, 24.8% notified)	Quantitative Data: - Seven percent of the total patients were children <15 years - Notification was significantly higher with microbiologically confirmed diagnoses - Notification was significantly lower in inpatients, children, and patients found through laboratory/pharmacy systems Qualitative Data: - Interviews described barriers to notification including diagnostic procedures and treatment, misconceptions about the notification process are common - Interviews described solutions including establishing more hospital systems for notifications
13	Yaqoob et al.	Cross-sectional study	2021	Pakistan	0–14	General	Non-NTP private facilities in 12 districts across Pakistan/NTP	97% (6332/ 6519)	Cases were less likely to be reported if private doctors started TB therapy themselves. Noted poor coordination between treatment centers and laboratories as potential cause for under-reporting. Noted inadequate counseling for patients with presumed TB and weak referral mechanisms.

*: Not described in paper, **: Unclear denominator for total cases. Abbreviations: TB, Tuberculosis; NTP, National Tuberculosis Program; TBIMS, Tuberculosis Information Management System; RNTCP, Revised National Tuberculosis Control Programme; DOTS, Directly Observed Treatment Shortcourse; TIBU, National TB surveillance system in Kenya; ETR.net, routine provincial electronic TB register.

**Table 3 tropicalmed-08-00300-t003:** Interventions described to close reporting gap.

Interventions Described to Close Reporting Gap									
#	Paper	Study Type	Year	Country (ies)	Age Range (Years)	TB Type	Hospital Type/ Reporting Body	Intervention	Effectiveness
14	Edginton et al.	Intervention study	2006	South Africa	0–55+, (0–14, 15–34)	General	Chris Hani Baragwanath hospital, (tertiary hospital)	Establishment of TB care center with registration within the hospital, death registration, education, and referrals	Increased patient registration with 94% of patients successfully referred to clinics.
15	Joshi et al.	Retrospective record review using routinely collected data	2015	Nepal	0–14	General	Seven of the 10 districts/ Nepal NTP	Intensified case finding detection with direct registration	Cases of childhood TB increased from 271 to 360 in the intervention districts (case registration rate from 18.2 to 24.2/100,000) compared with 97 to 113 in the control districts (case registration rate from 13.4 to 15.6/100,000).
16	du Preez et al.	Prospective cohort study	2018	South Africa	0–12	General	Tygerberg Hospital	Implementation of clinical surveillance along with previous laboratory surveillance, with the support of referral services between hospitals and local clinics	Clinical surveillance identified 237 (60%) of children that would have been missed by prior laboratory-based surveillance. Noted specific populations that were more likely to be identified by clinical surveillance including younger children, children with pulmonary TB, children with TB/HIV coinfection.
17	du Preez et al.	Prospective and retrospective cohort study	2020	South Africa	0–13	General	Tygerberg Hospital	Creation of a dedicated TB referral service within a pediatric ward	Successful reporting in 227/272 (84%) of children during the intervention period. Patients with culture-confirmed, drug-susceptible TB were more likely to be reported during the intervention period.
18	Shibu et al.	Intervention study	2020	India	0–65+ (0–14, 15–24)	General	8789 private doctors, 3438 chemists, and 985 laboratories	Pilot program “Private-practitioner interface agency (PPIA)” that engaged private providers by providing additional resources, monitoring quality, and supporting patients	PPIA notified 60,366 TB cases of tuberculosis in a 4-year period. The annual case notification rate per 100,000 population increased from 272 in 2013 pre-intervention to 416 in 2017.

Abbreviations: PPIA, private practitioner interface agency; NTP, National Tuberculosis Program.

## Data Availability

Not applicable.

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
