# Peer review of "Under-Reporting of Tuberculosis Disease among Children and Adolescents in Low and Middle-Income Countries: A Systematic Review"

_tropicalmed, 2023, doi:10.3390/tropicalmed8060300_

Round 1

Reviewer 1 Report

Dear colleagues,

The following items should be corrected:

-          The title of the review is not clear.

-          The structure of the abstract should be corrected and should include: background; objective; materials and methods; results; limitations, and conclusions.  Minimum 300-350 words .should be presented in the abstract.

-          The keywords not clear.

-          The aim of the study is not fully understandable and consistent with the title of the article – as it should be defined straightforwardly, otherwise upon review of the whole article it is still not clear what conclusions might be expected.

-    TB epidemiology data based on the WHO Reports 2020- 2022 should be included in the introduction. 

- Table 1 not clear.

- Colleagues should be introduced the link of PRISMA registration:

http://prisma-statement.org/Protocols/Registration

-  References are not up to date.

-          Таble 2 not clear. The data in the part «Trends in Reported Gap» should be corrected according to the aim of the study.

-          - Statistical data colleagues have not presented.

-           The results of the study is not clear.

-      Conclusion is not clear. In Conclusions chapter, general concepts and information are given and it is not clear practical relevance of this study. Please formulate the conclusions more clearly and precisely with practical relevance clarification.

The information in the chapter “Acknowledgments”,  “Funding”, “Declarations of Interest” have not presented.

All these concerns should be well addressed to consider this manuscript suitable for publication.

Reviewer 2 Report

Thanks for opportunity to review this well-written manuscript which reviews an important issue with current knowledge gap. The bias to under-reporting of the more severe (even fatal) disease is a particularly major challenge.

I only have two suggestions to consider.

Decentralisation of services for child and adoledcent TB is now a WHO recommendation (2022) and there is evidence (e.g. Zawedde-Muyanja S et al) and recent review by Courtney Yuen et al that this can lead to incerases in case detection and thereby case notification (potentially) if detection and decision to treat is coupled with notification from initiation of treatment. 

Is there a solution that couples initiation of treatment with mandatory notification at all levels of care for all types of TB? It used to be the case in Malawi that the TB drugs were kept in the district TB office so everyone was registered first in order to access treatment meaning that child TB notification data at least accurately reflected actual numbers diagnosed and treated (Harries AD et al. IJTLD)y. However, in those times, almost all cases were diagnosed as inpatients at a district hospital level. Decentralisation of detection and treatment to primary care level (which is often appropriate) may be more challenging to ensure notification. 

Round 2

Reviewer 1 Report

All edits  according my recommendations were done